# Testing Different Interpolation Methods Based on Single Beam Echosounder River Surveying. Case Study: Siret River

**Maxim Arseni \*, Mirela Voiculescu[ID], Lucian Puiu Georgescu, Catalina Iticescu and Adrian Rosu**

Faculty of Science and Environment, European Center of Excellence for the Environment, "Dunarea de Jos" University of Galati, 111, Domneasca Street, 800201 Galati, Romania; mirela.voiculescu@ugal.ro (M.V.); lucian.georgescu@ugal.ro (L.P.G.); catalina.iticescu@ugal.ro (C.I.); adrian.rosu@ugal.ro (A.R.)

\* Correspondence: maxim.arseni@ugal.ro; Tel.: +40-746407084

**Abstract:** Bathymetric measurements play an important role in assessing the sedimentation rate, deposition of pollutants, erosion rate, or monitoring of morphological changes in a river, lake, or accumulation basin. In order to create a coherent and continuous digital elevation model (DEM) of a river bed, various data interpolation methods are used, especially when single-beam bathymetric measurements do not cover the entire area and when there are areas which are not measured. Interpolation methods are based on numerical models applied to natural landscapes (e.g., meandering river) by taking into account various morphometric and morphologies and a wide range of scales. Obviously, each interpolation method, used in standard or customised form, yields different results. This study aims at testing four interpolation methods in order to determine the most appropriate method which will give an accurate description of the riverbed, based on single-beam bathymetric measurements. The four interpolation methods selected in the present research are: inverse distance weighting (IDW), radial basis function (RBF) with completely regularized spline (CRS) which uses deterministic interpolation, simple kriging (KRG) which is a geo-statistical method, and Topo to Raster (TopoR), a particular method specifically designed for creating continuous surfaces from various elevation points, contour, or polygon data, suitable for creating surfaces for hydrologic analysis. Digital elevation models (DEM's) were statistically analyzed and precision and errors were evaluated. The single-beam bathymetric measurements were made on the Siret River, between 0 and 35 km. To check and validate the methods, the experiment was repeated for five randomly selected cross-sections in a 1500 m section of the river. The results were then compared with the data extracted from each elevation model generated with each of the four interpolation methods. Our results show that: 1) TopoR is the most accurate technique, and 2) the two deterministic methods give large errors in bank areas, for the entire river channel and for the particular cross-sections.

**Keywords:** interpolation; river bathymetry; single-beam echosounder; river cross-section; Siret River

---

## 1. Introduction

The assessment of water bodies (lakes, rivers, etc.) properties makes use of knowledge regarding water quality, temperature, salinity, volumetric and depth measurements. The last contributes to the accurate maintenance of dredging, and the bathymetric mapping of supraglacial lakes, rivers, and streams, etc. The bathymetric models are the best solution to define the bottom surface of any type of lakes or rivers, and these models are obtained by means of depth measurements. The bathymetric light detection and ranging (LIDAR) scanning technology is currently the most applied surveying method for data collection, and it is applied to capture land and seafloor simultaneously, to create a detailed

3D elevation model along the coastline [1]. Although this is an effective and cost-efficient method, it cannot provide enough detail about the stream or river bathymetry, due to its inability to measure depth accurately where the water is cloudy or turbid [2]. In some cases, additional independent measurements are required to confirm LIDAR bathymetry performance, especially in deep, turbid waters or river cloudy waters, as mentioned e.g., by Kinzel [3] and Saylam [4]. Thus, if no bathymetric LIDAR is available, an accurate DEM (digital elevation model) for river beds can be obtained, when real-time kinematic (RTK) global positioning system (GPS) technologies are combined with multibeam or single-beam echosounders [5,6].

River bathymetry or the so-called "bed topography" plays a critical role in flow dynamics numerical modeling, in sediment transport modeling, geomorphologic modeling or in ecological assessments [7]. The best way to collect precise data for a river is by surveying it on a specific direction, namely on transversal cross-sectional paths [8,9]. The bathymetry measurement method is based on a boat-mounted single-beam or multibeam echosounder device combined with a global positioning system (GPS) with real-time kinematic (RTK) correction. Using this combination of measurements, a series of three-dimensional data (x, y, z) can be collected. The spatial resolution of collected data can be improved either by using a very expensive echosounder or by applying different methods of spatial interpolation in a post-processing step. Spatial interpolation of scattered data points has been an active research topic during the last decade since it has been applied in various fields. For example, Li and Heap [10] present a review of 53 comparative studies which assess the performance of 72 interpolation submethods and combined methods, used in different domains. Special attention is paid to the use of spatial interpolation techniques in hydrology, for interpolating scattered elevation data in order to create digital elevation models (DEM), which include the minor bed of river channel. According to Liffner [11], and Merwade [12] and Merwade [13], the DEM is affected directly by the type of interpolation methods used.

The present study aims at demonstrating the importance of choosing the appropriate spatial interpolation methods for interpolating river channel bathymetry data. The performance of four different interpolation techniques were analyzed by applying different statistical methods of performance assessment, such as: Minim value (MinV), maxim value (MaxV), mean error (ME), mean absolute error (MAE), mean square error (MSE), root mean square error (RMSE), root mean square standardized error (RMSSE) and standard deviation (SD) [13–18].

*Study Area*

The Siret hydrographic basin is located in the eastern part of Romania and it is the drainage basin of River Siret, which is the largest and most important tributary of the Danube [19]. The Siret River basin covers an area of 42,890 $km^2$ in Romania, 28.116 $km^2$ of which are managed by the Siret Water Directorate [20] under the name Siret Hydrographic Space (SHS). The study area covers a 35 km section upstream River Siret, along Galati–Sendreni–Independenta, i.e., from its confluence with the Danube, up to Independenta village. River Siret forms a natural border between Galati and Braila counties. Figure 1 shows an aerial map of the area under scrutiny.

The Siret riverbed material is composed of a combination of fine sand and silt [21]. The river bankfull along the studied area is approximately 100 m wide. Since no accurate bathymetric measurements have been made in this area [22], analyzing the bathymetry of this watercourse is scientifically relevant. Moreover, the measurements and three-dimensional information corresponding to this river section are of utmost contribution for enlarging the existing national databases.

The single-beam echosounder (SBES) is the most common instrument used in ports and in lake and river surveys, because it is an efficient, low cost and accessible instrument. The SBES uses acoustic depth measurements, which are based on measuring the elapsed time that an acoustic pulse takes in order to travel from a transducer to the waterway bottom and back. The general formula of the corrected depth is given by [23]:

$$d = \frac{1}{2}(v * t) + k + d_r. \tag{1}$$

where: $d$ = corrected depth; $v$ = average velocity of sound in the water; $t$ = time from transducer to bottom and back; $k$ = system index constant; $d_r$ = distance from reference water surface to transducer.

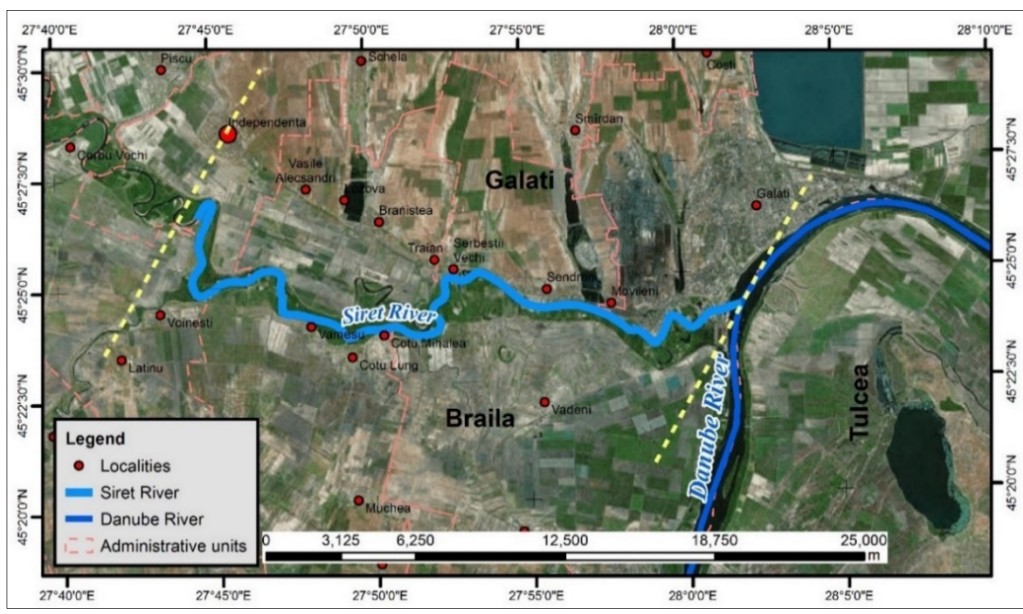

**Figure 1.** The study area located at the confluence of River Siret with the Danube. The 35 km area measured is positioned between the yellow lines.

## 2. Materials and Methods

### 2.1. Materials

The bathymetry data were collected by using a boat-mounted single-beam acoustic depth sounder (SBES) linked to a real-time kinematic (RTK) global positioning system (GPS), which can provide sub-decimeter accuracy for the surveyed points. Figure 2 shows several photos taken during data collection.

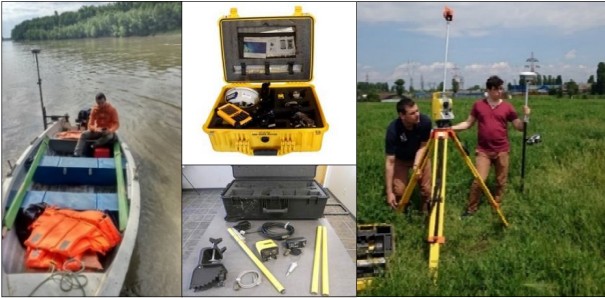

**Figure 2.** Left: Bathymetric depth measurements with SonarMite single-beam echosounder (SBES) (mounting method on a fiberglass rigid boat) linked to South S82-V RTK GNSS; middle: GNSS and SBES instruments and accessories; right: total station and South S82-V RTK GNSS used for topographic measurements.

Bathymetric depths were measured using an Ohmex SonarMite/BTX single-beam echosounder, with a 235 kHz frequency. The accuracy of the echo sounding equipment used to collect the bathymetric depth data is ±0.025 cm, with a ±4-degree beam spread, able to operate between 0.30 m to 75.00 m depth range (software limited). The sound velocity in water ranges between 1400 to 1600 m/s, but, when a sound velocity profiler (SVP) is not available, the echosounder uses a 1500 m/s average value. Depending on the water depth, the echosounder applies 3 to 6 Hz ultrasonic ping rate, with an output data range of 2 Hz [24].

The measurements were made in 5 days, within the interval 22.03.2017–01.04.2017. Each bathymetry point is characterized by the longitude and latitude obtained from the RTK GPS, and by the elevation (z), which is obtained by subtracting the water depth from the water surface elevation recorded at gauge station. The variation of the daily water level was obtained from the Sendreni gauge station. As Figure 3 shows, the water level during the measurement campaign varied between 435 and 460 cm.

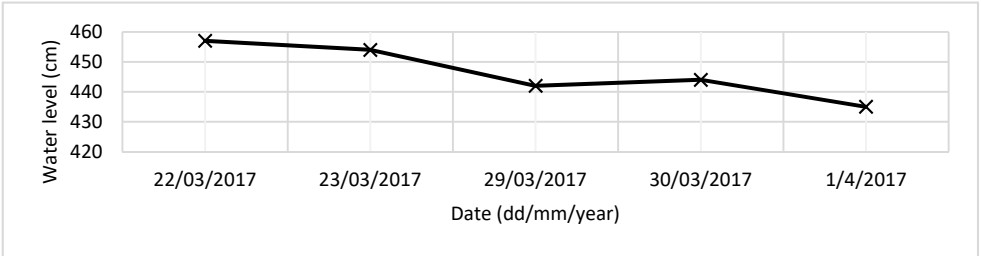

**Figure 3.** Water level at Sendreni gauge station recorded during the measurements campaign. Flood levels: C.A. (attention level) = 550; C.I. (flood level) = 600; C.P. (risk level) = 650.

The bathymetry data for the surveyed area (Figures 4 and 5) comprise a set of 216816 (x, y, z) points or approximately 45,000 points/km$^2$ measured along the river channel bed.

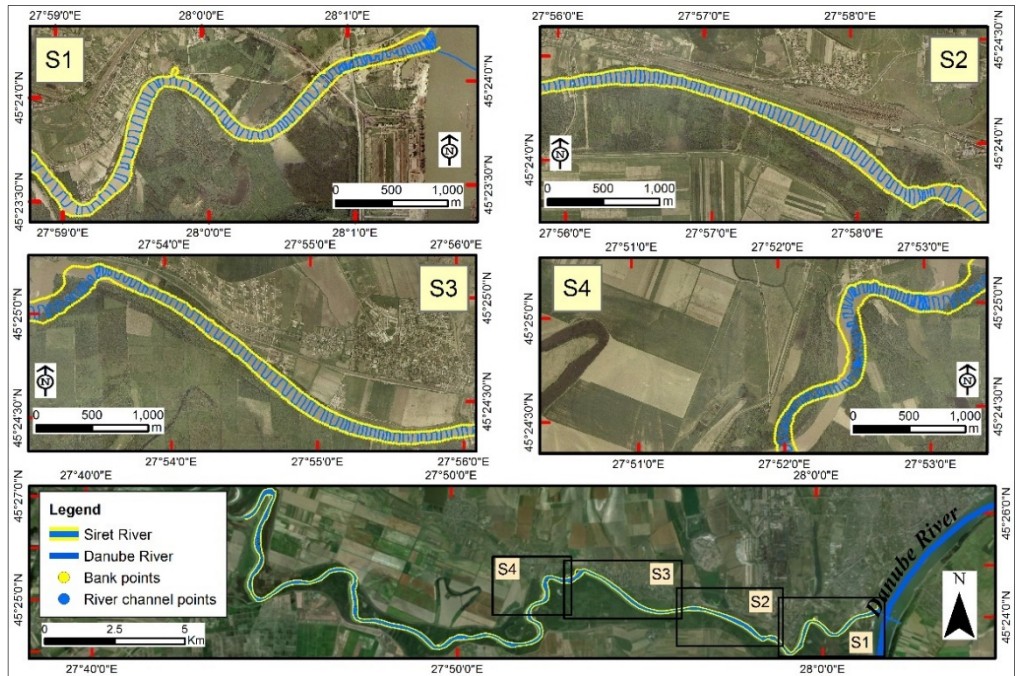

**Figure 4.** Top and middle: Ortopthomap views of sections S1 (downstream)–S4 (upstream); the blue line shows the trajectory of the boat; the river banks are marked in yellow. Bottom: ortopthomap view of the whole bathymetric survey area, with the first four sections.

Figures 4 and 5 present the 9 sections of the whole surveyed area. The total length of the surveyed path reached 103.22 km with an average boat speed of 1.75 knots (0.9 m/s). The distance between the cross-section ranges from 25 to 100 m. This distance depends on the linearity or sinuosity of the main river channel being larger in areas where the river track is straight, and smaller where the river is meandered. Since the water discharge rate is high in the case of River Siret (about 1000 m$^3$/s for a 4 m water level) [25], it was almost impossible to maintain equal distances between the cross-sections measured.

An ortopthomap with a 0.5 × 0.5 m planimetric resolution, produced in 2016, was used in order to represent the data in terms of geospatial location. This map was provided by the National Agency

for Cadastre and Land Registration of Romania (ANCPI). The bathymetric measurements and data processing observed the International Hydrographic Organization (IHO) S-44 regulation since Romania does not have a clearly defined national regulation for river bathymetric measurements. According to this regulation, the accuracy of bathymetric depth needs to comply with the requirements of the Special Order [26] which is used only for those areas where under-keel clearance is critical. The accuracy of depth data by IHO-S44 [26] regulation for this type of area is ±0.25 m, by applying the maximum allowable total vertical uncertainty (TVU), thus the results given by the SBES used for the present research article complies with the IHO S-44 regulations.

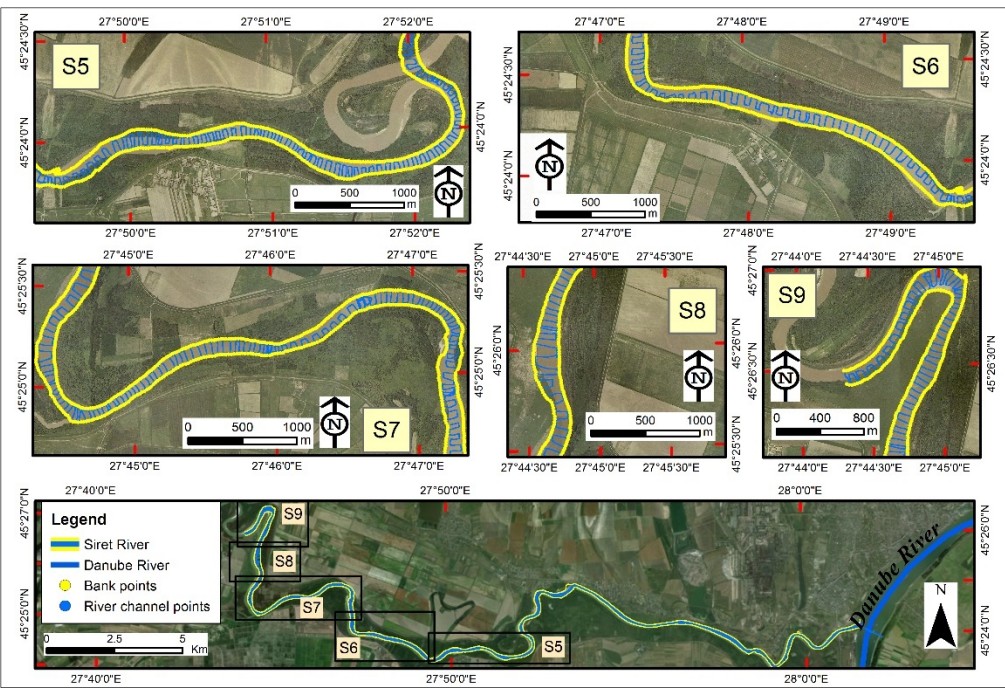

**Figure 5.** The same as in Figure 4, but for the last five sections of the bathymetric surveying: S5 (downstream)–S9 (upstream).

The Lat-Long coordinates and elevation (Z) of each point were measured with an RTK GPS – South S82-V equipment. The South S82-V is an RTK GNSS receiver, designed for precision acquisition data. This surveying instrument can receive GPS and satellite signals from GLONASS and GALILEO. The planimetric points were measured in the WGS84 coordinate system and the altimetric point in the Black Sea 1975 Constanta height Datum. By additionally using the RTK method the accuracy increases up to ±8 mm + 1 ppm RMS (Root Mean Square) for horizontal data surveying, and up to ±15 mm + 1 ppm RMS for vertical data surveying [27].

The bank points were measured by using a Trimble M5 surveying total station. The maximum distance between the points which mark the left and right river banks was 30 m. A combination of bathymetric and topographic measurements was used in order to achieve higher accuracy when the four interpolation methods were applied. The accuracy of the DEM depends directly on the number of measured points and it is crucial for flood mapping and hydraulic modeling.

To perform various functions, such as automated mapping functions, data management or spatial analysis function, the ArcGIS version 10.2., provided by ESRI (Environmental Systems Research Institute), was used. A GIS software populated with the correct data can help in the organization, interpretation, and communication of environmental data. For example, this particular GIS software has been used to create digital soil mapping, generated by kriging interpolation, in different urban areas [28]. Also, the GIS software was used [29] for comparing spatial interpolation methods in order to estimate the precipitation distribution. In our study, the GIS software was used for spatial data

exploration and surface generation, selecting different interpolation methods and statistical analyses which allow developing a DEM for riverbed channel from various data measured at discrete points.

*2.2. Methods*

The digital elevation model (DEM) is a widely used product and provides a three-dimensional representation (X, Y, Z) of the studied areas. According to Burrough [30], the term DEM can be defined as "a regular matrix representation of continuous variations of space relief units". This section describes the four different methods which are commonly used for interpolating scattered point data in order to obtain an accurate DEM for this section of River Siret. Calculations were performed by using the z values resulting from the bathymetric field measurements.

2.2.1. Inverse Distance Weighting (IDW)

IDW represents a deterministic spatial interpolation model, based on an assumption that the value at an unsampled point can be approximated by a weighted average of observed values within a circular search neighborhood [31]. The radius of the search circle is defined by the range of a fixed number of closest points. In most cases, the weights used for averaging the data value are a decreasing function of the distance between the sampled and unsampled points [32–34]. The general equation used by IDW in order to estimate a value at an unsampled point is given by:

$$z^* = \sum_{i=1}^{N} \lambda_i Z_i .$$  (2)

where $Z_i$ $(i = 1, 2, \ldots, N)$ are measured values at $N$ points, and the $\lambda_i$ is given by:

$$\lambda_i = (\frac{1}{d_i^p}) / (\sum_{i=1}^{N} \frac{1}{d_i^p})$$  (3)

where $d_i$ is the distance between *ith* sampled point ($z_i$) and the unsampled point ($z^*$), and $p$ is the exponent variable. A higher power exponent results in less influence from distant points, and vice versa. The standard power value is two.

2.2.2. Simple Kriging (KRG)

Kriging (KRG) is a method developed in the 1960s by the French mathematician Matheron [35] and exemplified in detail by Coburn [36]. This is a geostatistical method of interpolation which is very useful in different fields. Maps generated by this method have a very well-structured visual appearance, all based on irregular spatial data [35]. The general model of this method is based on a constant μ(s) for the data set and random errors ε(s) which have spatial dependence [37–40]:

$$Z(s) = \mu(s) + \varepsilon(s).$$  (4)

The prediction of value in a new cell position is done by summing the values of the weight of the multiplied points with the Z value of the data used in the interpolation [41]:

$$\hat{Z}(s_0) = \sum_{i=1}^{N} \lambda_i Z(s_i), \qquad \sum_{i=1}^{N} \lambda_i = 1$$  (5)

Being a very flexible method, it can be used by default or customized to match the data by specifying the nearest semi-variogram model. The semi-variogram consists of two components: The experimental semi-variogram (empirical) and the theoretical model of the semi-variogram [42].

The experimental semi-variogram results from the calculation of the variance of each measured point versus the other points used in spatialization [43]:

$$\hat{\lambda}\left(\overline{h_j}\right) = \frac{1}{2N_j} \sum_{i=1}^{N_j} [Z(s_i) - Z(s_i + h)]^2 \tag{6}$$

where: $N_j$ is a set of pairs of locations separated by the distance $h$; $\overline{h}$—represents the average of the distances between distinct pairs $N_j$.

Data analysis for KRG interpolation was performed using two particular cases of this method, namely: Simple kriging (SKRG) and universal kriging (UKRG).

The SKRG method involves determining the unknown value by estimating the weighted average values of the neighboring points [44]:

$$Z(s_0) = \sum_{i=1}^{N} \lambda_i Z(s_i) \pm \varepsilon. \tag{7}$$

where $Z(s_0)$ represents the estimated value, $Z(s_i)$ is the value of the neighboring points, $\lambda_i$ is the coefficients of weight, which satisfy the condition $\sum_{i=1}^{N} \lambda_i = 1$, and $\varepsilon$ is the default estimation error.

The UKRG method uses the assumption that the spatial variation of z is dependent on three components: a data set structure, a correlated random component, and a residual error. This is a hybrid method, where the spatial trend is measured by polynomial surfaces of different orders, derived globally or locally [45]:

$$\sum_{i=1}^{N} \lambda_i f_k(s_i) = f_k(s_0). \tag{8}$$

where $Z(s_0)$ represents the estimated value of the unknown variability, $\mu(s_0)$ is the deterministic function, and $\varepsilon(s_0)$ is random variation.

### 2.2.3. Radial Basis Function (RBF)

The radial basis function (RBF) is a interpolation method similar to the kriging family, but it does not benefit from the contribution of space spatial data analysis through the variogram [46]:

$$\hat{Z}(s_0) = \sum_{i=1}^{N} \omega_i \varphi(\| s_i - s_0 \|) + \omega_{n+1}. \tag{9}$$

where $\varphi(r)$ is the radial base function, $r = \| s_i - s_0 \|$ is the radial distance between the point for which a new $s_0$ value is calculated and the points with measured values $s_i$, and $\omega$ symbolizes the weights to be estimated.

Basic radial functions are often used to create neural networks. This method is used to calculate surfaces composed of a very large number of points [47]. According to Dumitrescu [48], the most applied RBF functions are:

1. Multiquadric function (MQ):

$$\varphi(r) = \left(r^2 + \sigma^2\right)^{1/2}. \tag{10}$$

2. Thin-plate spline (TPS):

$$\varphi(r) = (\sigma + r)^2 \ln(\sigma + r). \tag{11}$$

3. Spline with tension (ST):

$$\varphi(r) = \ln \sigma * \frac{r}{2} + K_0 \sigma * r + C_e \tag{12}$$

where $K_0(x)$ is the modified Bessel function.

4.  Completely regularized spline (CRS):

$$\theta(r) = -\sum_{n=1}^{\infty} \frac{(-1)^n (\sigma * r)^{2n}}{n! n} = \ln\left(\sigma * \frac{r}{2}\right)^2 + E_1\left(\sigma * \frac{r}{2}\right)^2 + C_e. \tag{13}$$

where $E_1(x)$ is the exponential integration function, and $C_e$ is the Euler constant.

### 2.2.4. Topo to Raster Interpolation (TopoR)

The Topo to Raster interpolation method (TopoR) is specifically designed for creating hydrologically correct digital elevation models (DEMs) [49]. It is based on the ANUDEM program developed by Michael Hutchinson [50–53]. This method was designed to take advantage of all types of possible input data which typically characterize the landforms (elevation points, contour lines, stream centerline, sink, boundary, lake, cliff, coast) [54].

The Topo to Raster interpolation was developed by the ANU (Australian National University), but this method is often used in many countries and research papers [55,56]. It uses a finite differential interpolation technique and it is optimized to streamline the local interpolation computational method such as IDW interpolation, without losing the continuity of surfaces obtained by the kriging or RBF Spline interpolation methods. The difference is defined as the first and second degree of partial derivation $f$ of the interpolation method described by the following Equations [56]:

$$J_1(f) = \int \left(f_x^2 + f_y^2\right) dxdy. \tag{14}$$

$$J_2(f) = \int \left(f_{xx}^2 + f_{xy}^2 + f_{yy}^2\right) dxdy. \tag{15}$$

$J_1$ and $J_2$ must be minimized in order to eliminate the maximum and minimum peak effect (excessively smooth or peaked surface) and to obtain a realistic surface terrain. If only $J_2$ is minimized, a very smooth surface is obtained, and vice versa, if only $J_1$ is minimized, maximum and minimum peaks occur [57]. Hutchinson [50] suggests applying a roughness penalty by taking into account the cell resolution so as to reduce this effect:

$$J(f) = 0.5h^{-2} J_1(f) + J_2(f). \tag{16}$$

Essentially, TopoR is a combined interpolation method which uses a discrete technique based on spline polynomial functions of degree $m$ and smoothness $k$, where the roughness can be modified to allow the generation of DEM with steep changes in the field, such as areas influenced by depths of currents, ridges or cliffs. These landforms are also called local maximums or local minimums. In the present research, point elevation features were used.

## 3. Results and Discussion

### 3.1. Block Data Performance Analysis

After collecting and validating topo-bathymetric data, digital models were generated (Figure 6), by using the four interpolation methods described in the previous section.

Figure 6a–d presents the four digital elevation models: (a) inverse distance weighting (IDW) interpolation; (b) kriging (KRG) interpolation; (c) radial basis function (RBF) interpolation with completely regularized spline (CRS) method; (d) Topo to Raster interpolation method, for a selected river section of 1500 m. The spatial resolution of $5 \times 5$ m was used for all DEMs.

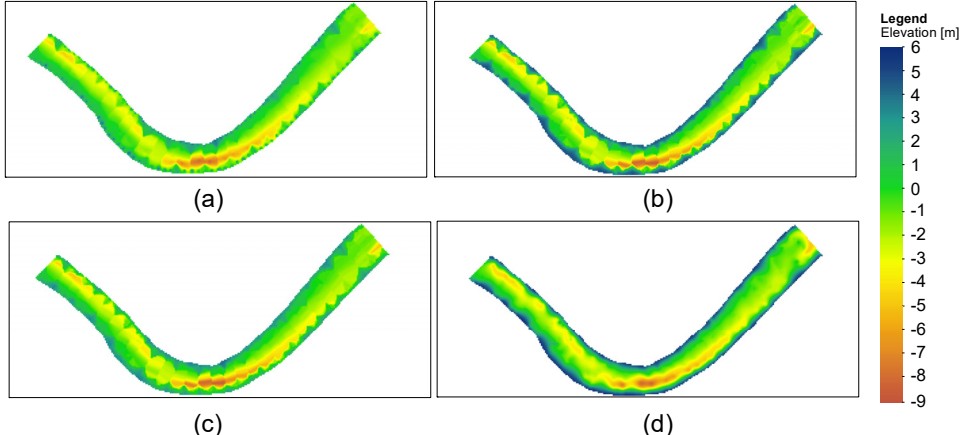

**Figure 6.** Digital elevation model (DEM) of a 1500 m river section of the main channel obtained by using the 4 interpolation methods: (**a**) inverse distance weighting (IDW); (**b**) kriging (KRG); (**c**) radial basis function with completely regularized spline (RBF); (**d**) Topo to Raster.

All interpolation methods were run by using Spatial Analyst Tools from ArcGIS 10.2. For the simple kriging method, a single spherical variogram model was fitted to the sample variogram of all data by varying different nugget, sill, and range to fit the variogram model. Special attention was paid to matching the slope for the first several reliable lags. Due to this fact a spherical semivariogram, with Nugget = 0.012, Sill = 1.15, Range = 106.44, and Lag Size = 13.30 was adopted.

In the case of the TopoR interpolation method, the differences in elevation ranges were least obvious than those resulting from the IDW and RBF interpolation methods. The TopoR DEM has a much smoother and flatter graphic representation, with lower sinuosity elements.

The basic statistical value which defines the dispersion of the frequency distribution of deviations between the measured and unmeasured interpolated values is the mean-standard deviation (mean SD). Figure 7 shows the Gauss curve representation (a) and boxplot chart (b) for each model, which illustrate how closely the created model predicts the measured values. The Gauss distribution shown in Figure 7a proves that the narrowest line is associated to the TopoR method. A good model will be described by a narrow error distribution and a small, close to zero, SD. Both plots in Figure 7a,b suggest that TopoR is the best model, since the associated error histogram is the narrowest and the mean SD is the smallest.

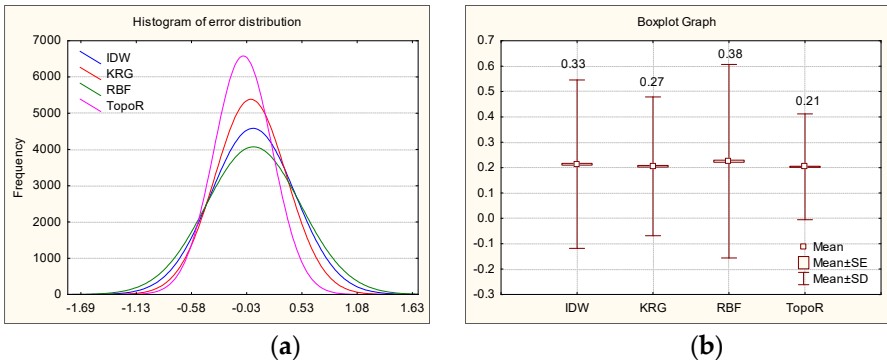

**Figure 7.** Statistical results of model fit: (**a**) The frequency of error distribution for each of the four models: IDW (blue), KRG (red), RBF (green), TopoR (magenta); (**b**) Mean standard deviation between measured and interpolated z-values for each of the four models. IDW, inverse distance weighting; KRG, simple kriging; RBF, radial basis function.

For a fair comparison of the interpolated points on the chosen section, the same number of points and the 5 m × 5 m spatial resolution of the raster were used. In order to assess the accuracy of each interpolation methods with the SBES measurements, some statistical tests whose results are shown in

Tables 1 and 2, were performed. Table 1 describes the results of obtained mean elevation and variance for all points along the 1500 m river section, and the differences between the depth measurements (SBES) with the four interpolators by using a two-tailed t-test. Values of correlation coefficients and results of the regression analysis are shown in Table 2.

**Table 1.** Report on the results from T-test to test the significance of the compared method.

| Results / Method | Mean | Variance | Count | T-stat | P(T < = t) Two-Tail |
|---|---|---|---|---|---|
| | | | | **T-Test** | |
| SBES | −1.060 | 4.170 | 8709 | - | - |
| IDW | −1.087 | 3.724 | 8709 | 0.917 | 0.358 |
| KRG | −1.065 | 3.963 | 8709 | 0.174 | 0.861 |
| RBF | −1.091 | 3.708 | 8709 | 1.062 | 0.288 |
| TopoR | −0.983 | 4.119 | 8709 | −2.271 | 0.023 |

**Table 2.** Results of correlation between SBES and the DEMs obtained by using interpolation methods for the 1500m river section (left column) and regression (right column) for each of the four DEMs.

| Method | Correlation Test | Regression Analysis | | |
|---|---|---|---|---|
| | Pearson Coefficient | R2 | SD | P-Value |
| IDW | 0.979 | 0.959 | 0.414 | $1.53 \times 10^{-38}$ |
| KRG | 0.984 | 0.969 | 0.356 | $4.15 \times 10^{-4}$ |
| RBF | 0.973 | 0.947 | 0.468 | $1.54 \times 10^{-30}$ |
| TopoR | 0.988 | 0.973 | 0.306 | $6.54 \times 10^{-109}$ |

Scatterplots of interpolated (Vi) and measured values (Vm) can be seen in Figure 8a–d, together with the corresponding correlation coefficient. The four methods give reliable results for most elevation values, except for the interval 4–6 m. Correlation coefficients are close to 1 for all methods (see Table 2); however, not all methods give similar results. Table 2 shows that the RBF method has the lowest performance, with the smallest regression coefficient, $R^2 = 0.947$, and a correlation coefficient of 0.973. This is in line with Arseni [18] and Rosu [58], who found a similar result.

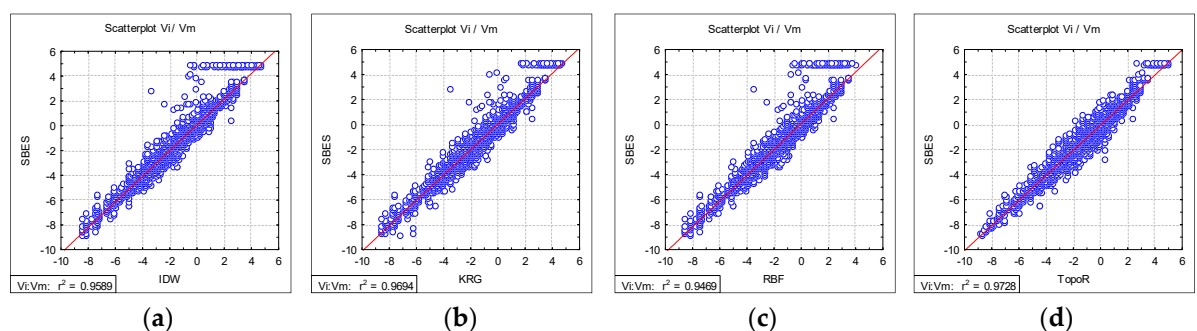

**Figure 8.** The linear regression for the four interpolation methods: (**a**) IDW; (**b**) KRG; (**c**) RBF; (**d**) TopoR.

The TopoR interpolation method gives the best results, with the highest values for both regression and correlation coefficients ($R^2 = 0.973$ and 0.988). Also, the DEM produced with TopoR has the lowest number of points within the problematic sector of 4–6 m. This is in line with Arseni [22], who found a similar result.

*3.2. Local Cross-Section Performance Analysis*

In order understand better the qualities of each of the four interpolation methods, the individual behaviour of the models for five cross-sections chosen for validation was also analyzed. Thus, five

random cross-sections were chosen for repeated surveying and data validation from the 1500 m studied area (Figure 9).

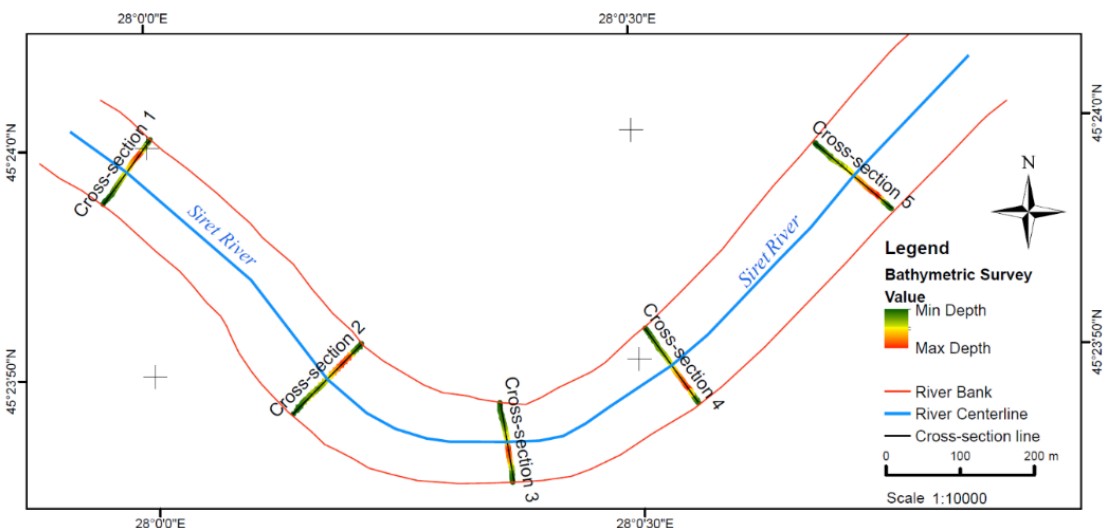

**Figure 9.** Locations of the five-cross-section measured with SBES technique.

The SBES profile was obtained from measurements by using the linear method (cross-sectional survey). The four modeled profiles were obtained by extracting each point value from the generated DEMs. Each cross-section surveyed by SBES bathymetric measurements was compared with the values extracted from each DEM characteristic of each interpolation method. The differences between each model and measurements for each cross-section is shown in Figure 10. Detailed information about the five statistical parameters (variables) for models and measurements are presented in Table A1 (Appendix B): minimum value (MIN), maximum value (MAX), mean value (MEAN), median (MEDIAN), and standard deviation (SD), calculated by using the ArcGIS 10.4 software. These parameters were used to assess the success of the data analysis.

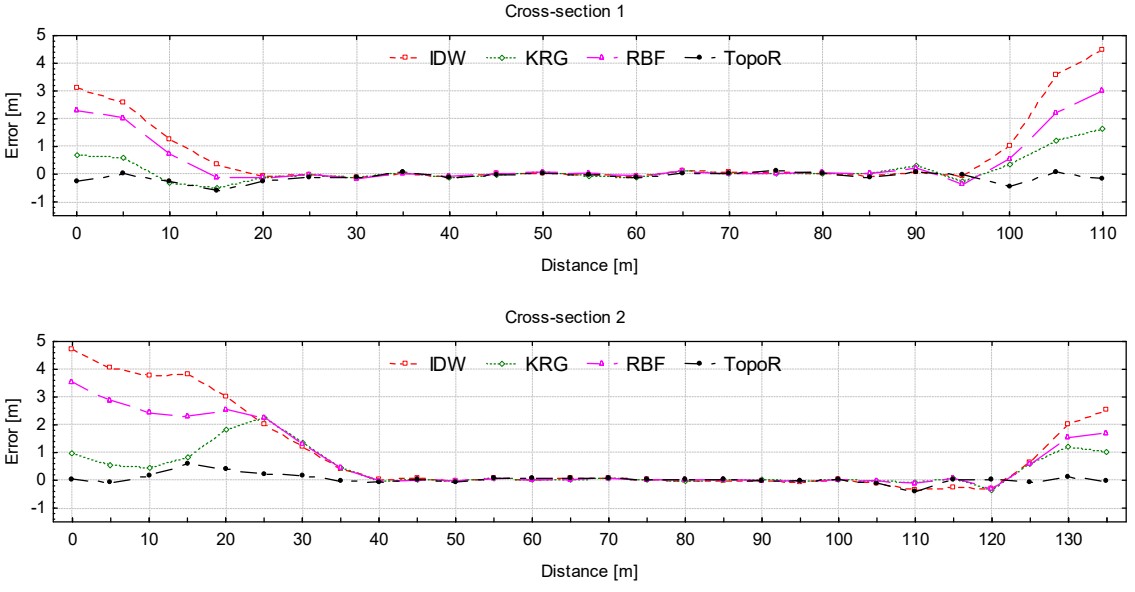

**Figure 10.** *Cont.*

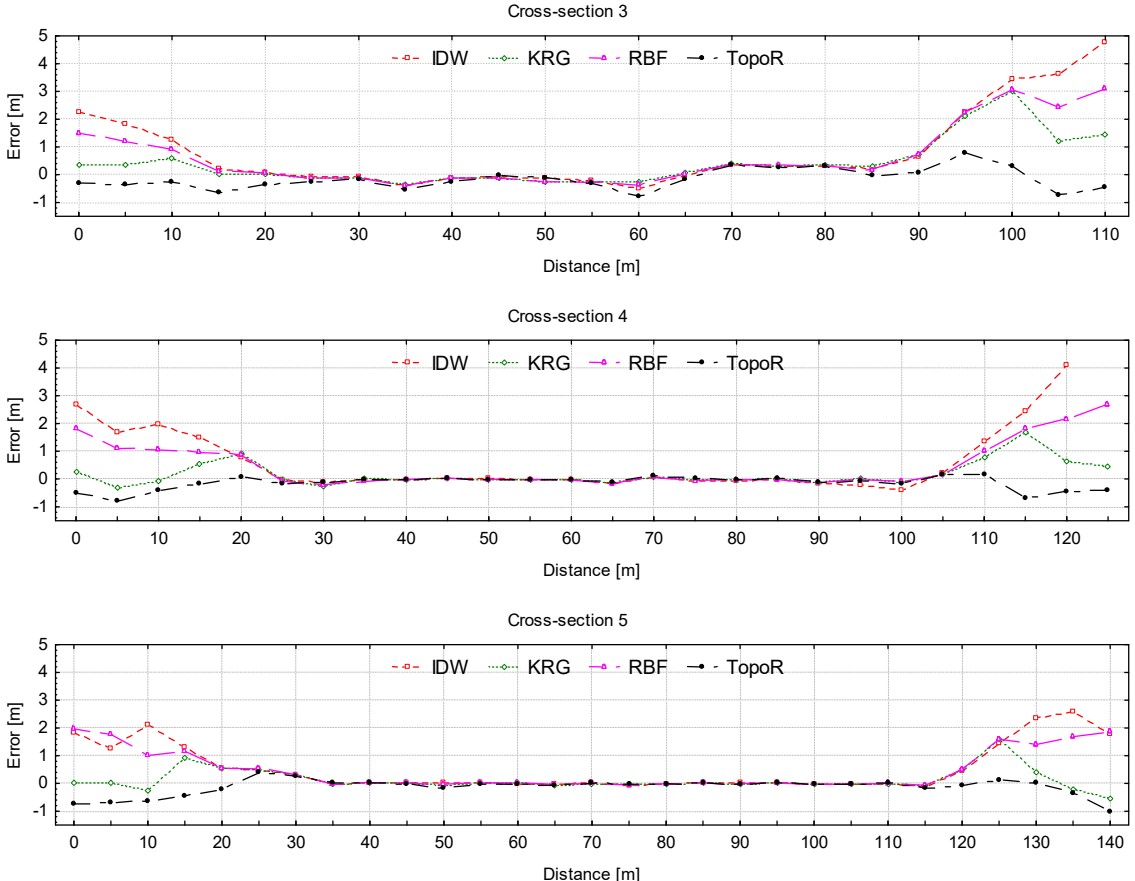

**Figure 10.** Difference between measurements (by SBES) and interpolated elevation values from each DEM model (SBES–DEM): IDW—red line, KRG—green line, RBF—magenta line, TopoR—black line; negative values correspond to overestimating the elevation.

The differences between SBES measurements and interpolated elevation value, for each of the five section, are shown in Figure 10 in order to assess, in a direct manner, the performance of each model. Figure 10 shows that all models reproduce very well the river bed except for two regions of about 20–30 m from the river banks, which are basically regions of small depths or positive elevation (as seen in Figure A1/Appendix A).

The deviation increases when approaching both banks for all models. TopoR slightly overestimates the elevation for all sections, while IDW and RBF underestimate the elevation at the shores. The KRG model has an irregular behavior, with values which are either larger or smaller than measurements. Obviously the TopoR model is the best, since the deviations from measurements are the smallest (1 m). On the other hand, IDW has the worst performance, with differences between the model and the field measurements of more than 5 m (5.35 m for cross-Section 4, close to right bank of the river flow). It seems that the IDW, KRG and RBF methods are not reliable as long as the river banks have steep slopes.

Further, all data have been normalized by the value of the elevation ($Z$), which represents the elevation of the river bed channel. Normalization was performed by recalculating the $Z$ values between 0 and 1, using the following formula:

$$Z_i = \frac{X_i - \min(X)}{\max(X) - \min(X)} \tag{17}$$

where $X = (X_1, X_2, \ldots, X_n)$ and $Z_i$ represent the $i$th normalized data.

This type of normalization is also called "feature scaling", or "unity-based normalization". Another type of normalization (like normalization by dividing to maximum value) cannot provide scaled data between 0 and 1, considering the negative elevation values.

The normalized profiles obtained from SBES measurements and the four interpolation methods for each cross-section are shown in Figure 11.

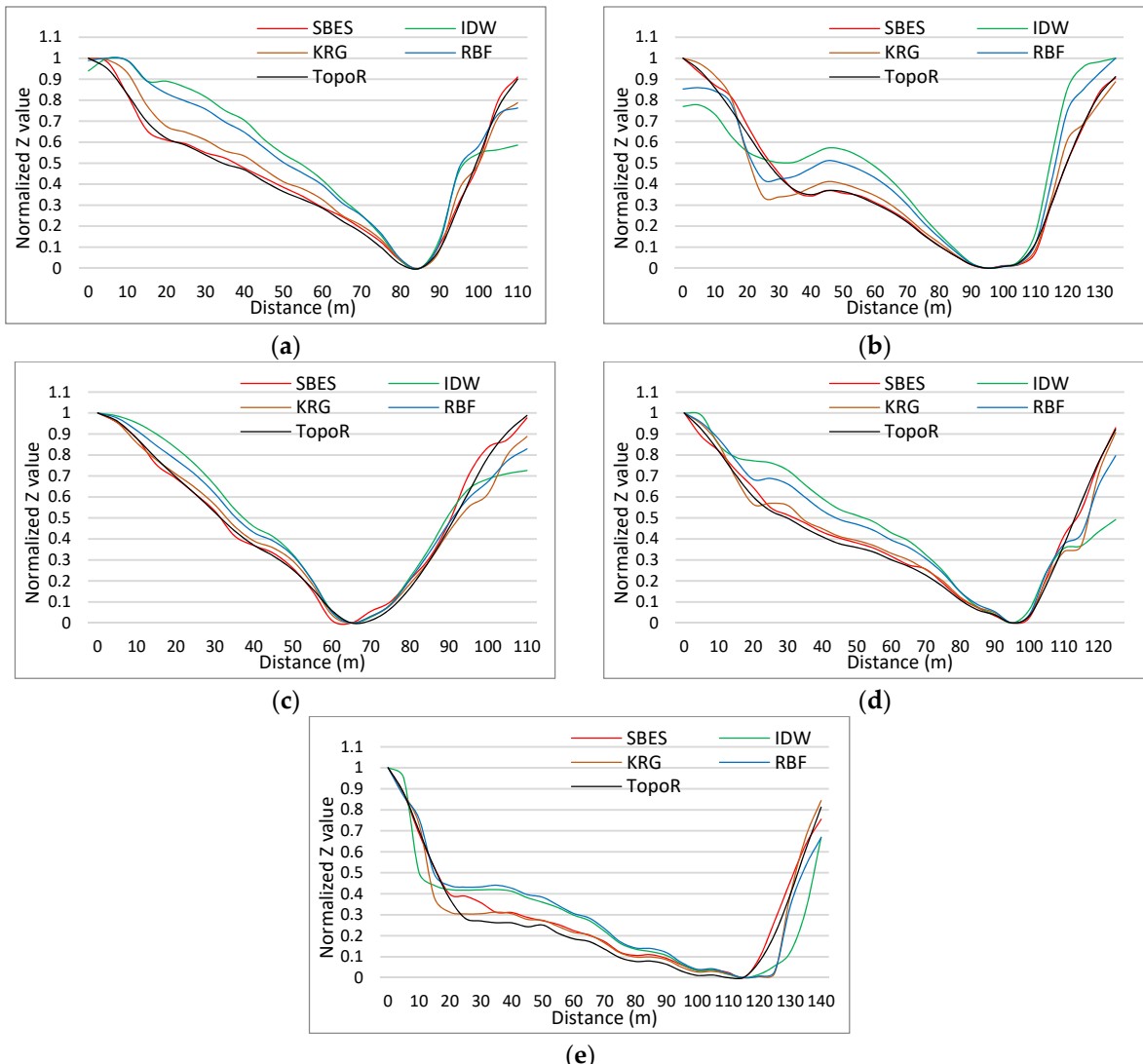

**Figure 11.** Comparison between the SBES normalized profile (red) and normalized profiles extracted from DEM (orange: KRG, black: TopoR, green: IDW and blue: RBF), for five cross-sections for River Siret: (**a**) Cross-Section 1; (**b**) Cross-Section 2; (**c**) Cross-Section 3; (**d**) Cross-Section 4; (**e**) Cross-Section 5.

Figure 11 shows that each transversal profile obtained by interpolation methods reasonably describes the measured profile of the selected cross-sections of the river bed channel. The good performance of the TopoR DEM (black line) is illustrated in Figure 11, since the corresponding profile is very close (sometimes even superimposed) to the profile given by SBES bathymetric measurements (red line). The other three models give sometimes erroneous results, especially in the area close to the right and left banks, as shown in Figure 10. This can be also seen in Appendix A, Figure A1, where plots of normalized profiles are compared to real profiles. This is in accordance with the observed higher dispersion seen for low depths (0–5 m) in Figure 8.

Figure 12 shows results of the regression analysis for each of the four methods on SBES measurements (normalized values).

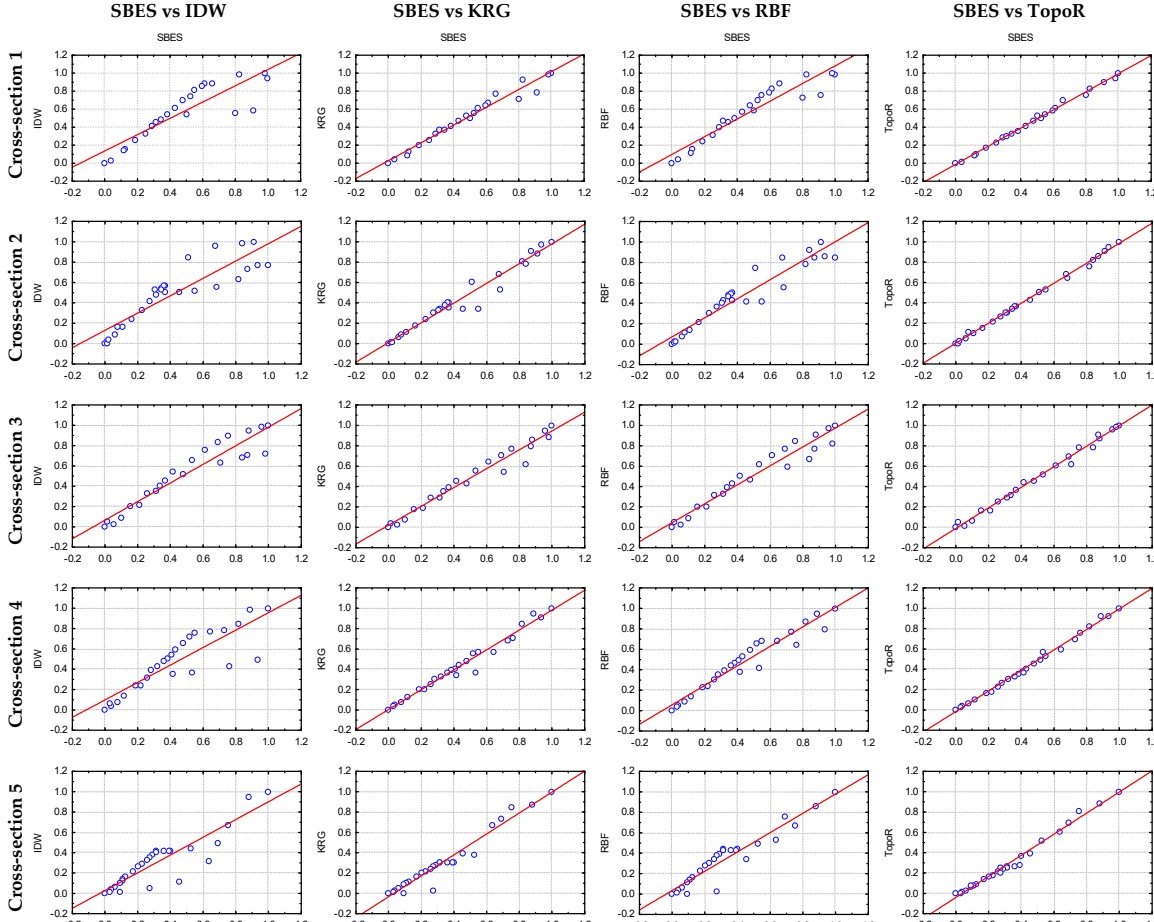

**Figure 12.** Simple linear regression of the four DEM-based normalized profile on the normalized SBES profile (left to right: IDW, KRG, RBF, TopoR) for each selected cross-section.

The regression analysis shows that the IDW method has a diffuse data dispersion in all 5 cross-sections. Thus, this is the most inaccurate method. The other two methods, i.e., KRG and RBF, have less regular behavior, with a higher data dispersion for some sections. Again, TopoR is the least dispersive method, whose results are almost perfectly linearly dependent on measurements for all cross-sections.

The statistical analysis was performed by using the Pearson correlation factor (R) and root mean square error (RMSE). The results obtained are shown in Figure 13. The root mean square error (RMSE) or so-called root mean square deviation (RMSD) measures the deviation of output data from the initial input data [59,60]. A small RMSE together with a high R indicate a good fit of the model. Obviously, TopoR has the best performance, closely followed by the KRG method, while IDW is the poorest interpolation method in all cases.

Our results are in line with Curebal's [61] conclusions for the Keçidere basin, or for models of the Aswan high dam reservoir storage capacity and morphology development [62], which both show that TopoR is the most accurate method for obtaining hydrologically correct DEM. The IDW and KRG methods are reported as being inaccurate when analyzing river bed topography by Goff [63] and Merwade [13]. However, sometimes the kriging (KRG) method estimates the elevation better than IDW [64]. Also, Šiljeg [57] shows that that the results based on TopoR has no significant differences (small errors) in comparison with the aero-photogrammetric measurements (which made up his base model). However, Šiljeg [57] reports that TopoR is not recommended for the micro-level analysis of, e.g., rocks or specific micro-landforms. A complex analysis of many interpolation methods for creating DEMs in the Belgorod region of Podsadneea [65], shows that TopoR is the most accurate method.

However, despite the accuracy of interpolation techniques for generating accurate DEM assessed by several studies, there are still no consistent findings about the performances of the spatial interpolators for river bathymetry [66].

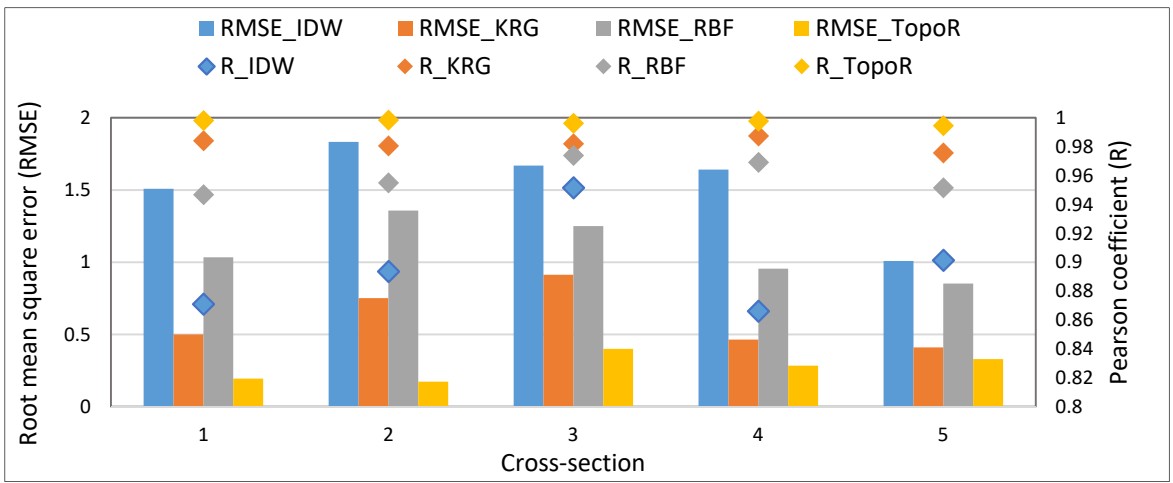

**Figure 13.** Root mean square error (RMSE) bar, left axis, and the Pearson correlation coefficient (R) diamonds, right axis, for modeled values for the selected cross-sections: blue: IDW, orange: KRG, grey: RPF, yellow: TopoR. Please note that the best fit is given by high R and small RMSE.

## 4. Conclusions

The creation of the digital elevation model of an area depends on the interpolation method. A good DEM helps calculating the plan and volumetric areas of the river more precisely; thus, DEM can be very useful for evaluating any morphometric changes. This study aimed at identifying the best interpolation method which can give the most accurate representation of a river bed from single-beam river bathymetry measurements.

The bathymetric survey was carried out the on section of River Siret and was based on single-beam echosounder combined with an RTK GPS system. The digital elevation bathymetric models created by single-beam measurement technique depend on the interpolation methods used. In order to assess the accuracy of the interpolated surface, commonly available interpolation methods such as IDW, KRG, RBF and TopoR [61] were used. The Mean SD was one of the statistical indicators which was used to assess the interpolation accuracy. There were no differences between the measured and unmeasured interpolated values in the case of accurate interpolation (e.g., IDW). One can only guess that these differences were determined by cross-validation. The cross-validation of block data showed that the TopoR interpolation has the smallest standard deviation (0.306 m). The coefficients of the determination $R^2$ between the predicted and the measured values were $R_{IDW}^2 = 0.959$, $R_{RBF}^2 = 0.947$, $R_{KRG}^2 = 0.969$ and $R_{TopoR}^2 = 0.973$, confirming that TopoR is the most accurate method. Differences were also observed in the case of the digital elevation model representation. The TopoR interpolation method described an attenuated and smoothed DEM (Appendix A, Figures A2–A5).

The DEM obtained by using the four interpolation methods were compared with the SBES measurements for the five cross-sections of the river in order to check the accuracy of each method. The TopoR interpolation profile is almost similar to the SBES measurements profile. The profiles obtained by using the IDW, KRG, and RBF methods had sizeable differences as compared to the measured profile, especially in the areas close to the river banks.

The statistical results show that all DEMs are statistically significant, regardless of the method; however, the TopoR interpolation method is the best, with the smallest RMSE and the highest Pearson correlation coefficient, when comparing the modeled and measured depth data. The statistical parameters are: $RMSE_{TopoR} = 0.276$ m and $R_{TopoR} = 0.997$ m, against the IDW method, which has a mean $RMSE_{IDW} = 1.532$ m and $R_{IDW} = 0.897$ m.

Thus, all results show that the TopoR method is the most accurate interpolation method to be used when creating a DEM, for a given number of points and grid.

Ideas of a future study have emerged from the analysis of these tests, and one refers to adding bathymetric longitudinal measurements (along the river channel or perpendicular to cross-sections), thus creating a more regular measurement grid. This may improve the analysis of interpolation methods and will also lead to a higher accuracy. The number of points and pixel size (spatial resolution) is another important parameter worth taking into account. The investigation of the effect of these on digital elevation models is presently on the way.

**Author Contributions:** Maxim Arseni and Adrian Rosu performed the measurements, establish the methodology, validated the data and analyzed the data; Adrian Rosu designed the figures; Lucian Puiu Georgescu and Catalina Iticescu supervised the progress of teamwork; Maxim Arseni did the statistical analysis and, together with Mirela Voiculescu, prepared the manuscript. All authors commented on the original manuscript.

**Funding:** This work was partially co-financed by the European Social Fund, through The Human Capital Operational Programme 2014–2020 (POCU), Romania, POCU/380/6/13.

**Acknowledgments:** Arseni Maxim's and Rosu Adrian's research work was supported by the project "ANTREPRENORDOC", Contract no. 36355/23.05.2019, financed by The Human Capital Operational Programme 2014–2020 (POCU), Romania, and the work of Voiculescu Mirela, Iticescu Catalina and Georgescu Puiu Lucian was supported by the project "EXPERT", financed by the Romanian Ministry of Research and Innovation, Contract no. 14PFE/17.10.2018.

**Conflicts of Interest:** The authors declare no conflict of interest.

## Appendix A

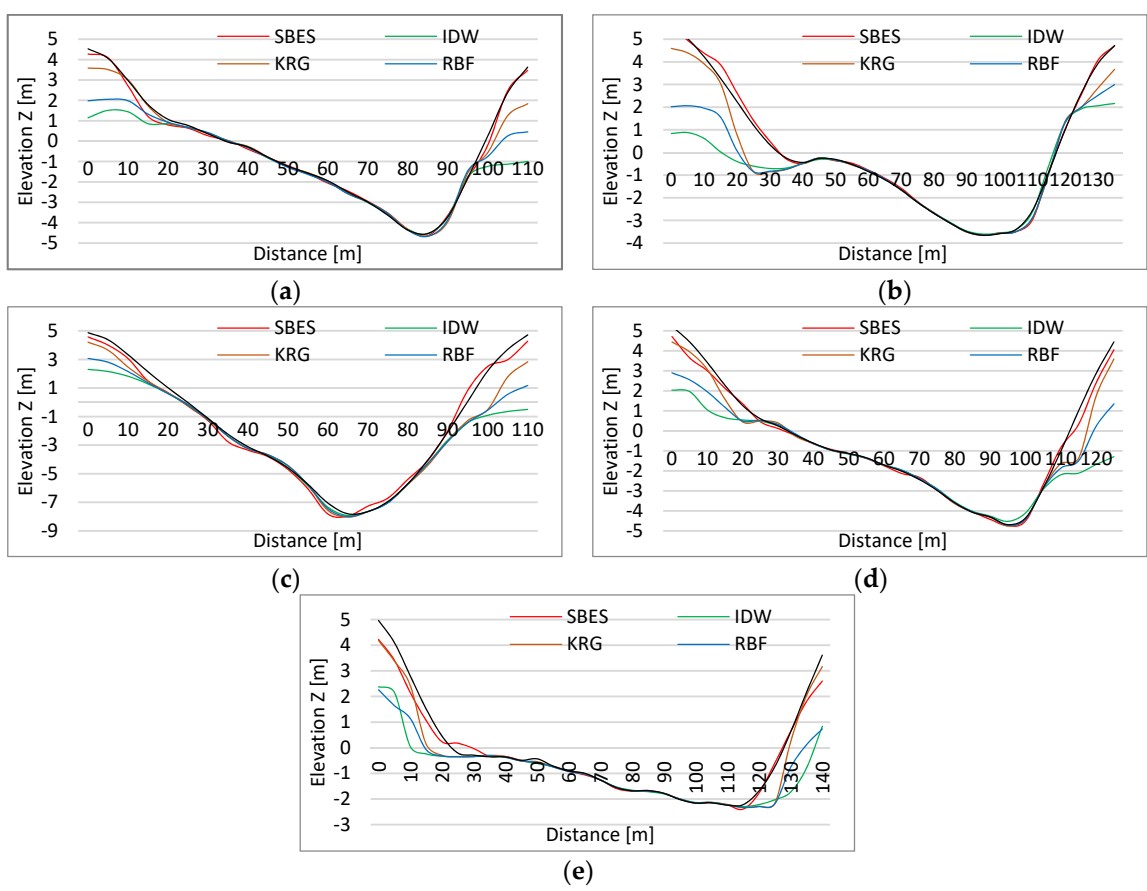

**Figure A1.** Profile shape from right to left bank of SBES and extracted from DEM profiles for the Siret River: (**a**) Cross-Section 1; (**b**) Cross-Section 2; (**c**) Cross-Section 3; (**d**) Cross-Section 4; (**e**) Cross-Section 5.

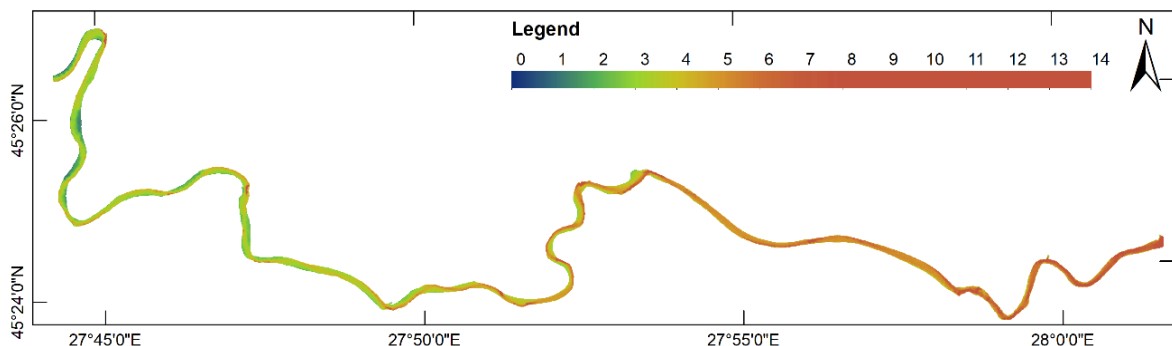

**Figure A2.** River Siret depth map obtained by using the IDW interpolation method.

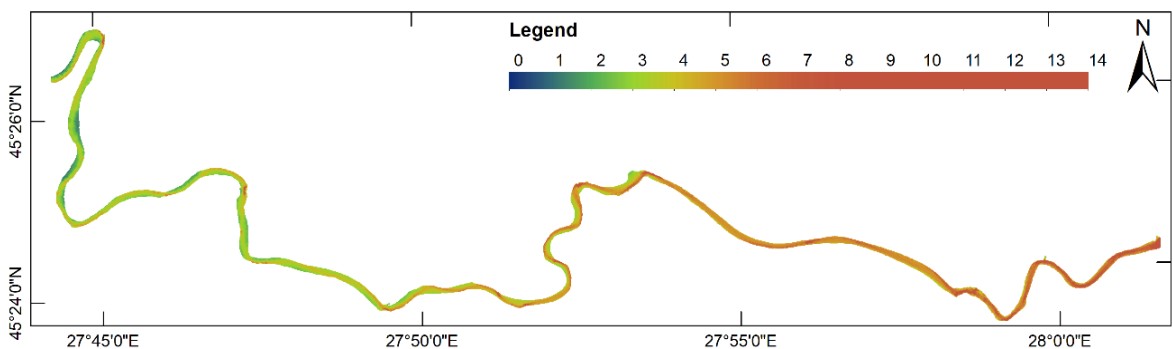

**Figure A3.** River Siret depth map obtained by using the KRG interpolation method.

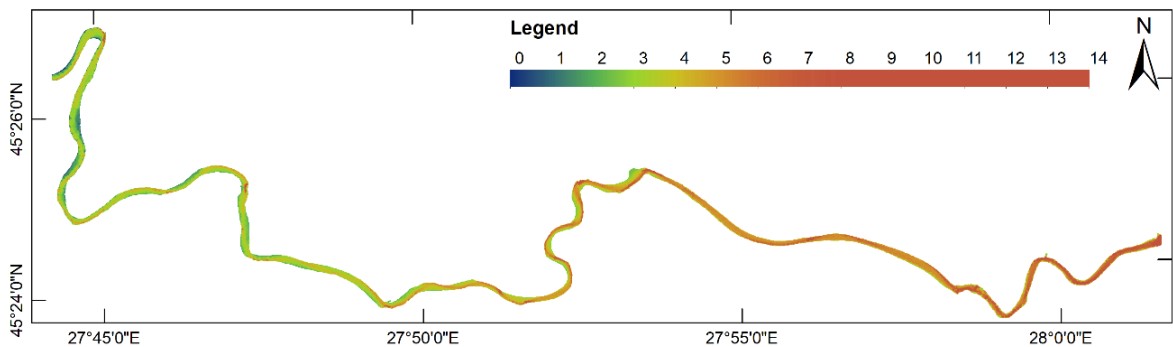

**Figure A4.** River Siret depth map obtained by using the RBF interpolation method.

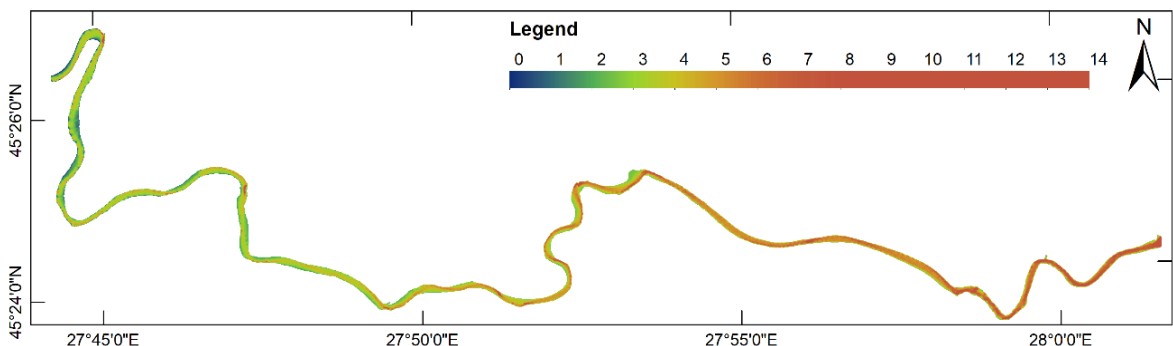

**Figure A5.** River Siret depth map obtained by using the TopoR interpolation method.

**Appendix B**

**Table A1.** Statistics of model fits as given by the ArcGIS report.

| Method / Results | MIN | MAX | MEAN | MEDIAN | SD | Number of Measured Points | Width [m] |
|---|---|---|---|---|---|---|---|
| **Cross-Section 1** | | | | | | | |
| SBES | −4.636 | 4.272 | −0.420 | −0.396 | 2.629 | | |
| IDW | −4.546 | 1.512 | −1.128 | −1.129 | 1.858 | | |
| KRG | −4.653 | 3.583 | −0.573 | −0.562 | 2.445 | 237 | 109 |
| RBF | −4.668 | 2.052 | −0.880 | −0.758 | 2.063 | | |
| TopoR | −4.530 | 4.528 | −0.330 | −0.287 | 2.702 | | |
| **Cross-Section 2** | | | | | | | |
| SBES | −3.661 | 5.567 | 0.156 | −0.778 | 2.928 | | |
| IDW | −3.605 | 2.163 | −0.824 | −0.706 | 1.745 | | |
| KRG | −3.652 | 4.596 | −0.255 | −0.853 | 2.585 | 190 | 128 |
| RBF | −3.649 | 2.996 | −0.611 | −0.845 | 2.056 | | |
| TopoR | −3.655 | 5.525 | 0.111 | −0.844 | 2.861 | | |
| **Cross-Section 3** | | | | | | | |
| SBES | −7.973 | 4.569 | −1.700 | −1.989 | 4.172 | | |
| IDW | −7.927 | 2.304 | −2.568 | −2.366 | 3.277 | | |
| KRG | −8.038 | 4.208 | −2.119 | −2.389 | 3.833 | 174 | 109 |
| RBF | −8.000 | 3.070 | −2.347 | −2.364 | 3.513 | | |
| TopoR | −7.806 | 4.868 | −1.545 | −2.085 | 4.267 | | |
| **Cross-Section 4** | | | | | | | |
| SBES | −4.748 | 4.713 | −0.624 | −0.879 | 2.755 | | |
| IDW | −4.526 | 2.040 | −1.423 | −1.524 | 1.894 | | |
| KRG | −4.751 | 4.444 | −0.797 | −1.254 | 2.653 | 189 | 118 |
| RBF | −4.719 | 2.901 | −1.118 | −1.252 | 2.184 | | |
| TopoR | −4.688 | 5.223 | −0.481 | −0.982 | 2.922 | | |
| **Cross-Section 5** | | | | | | | |
| SBES | −2.394 | 4.224 | −0.326 | −0.599 | 1.756 | | |
| IDW | −2.294 | 2.379 | −0.890 | −0.901 | 1.202 | | |
| KRG | −2.325 | 4.180 | −0.448 | −0.737 | 1.821 | 197 | 137 |
| RBF | −2.328 | 2.260 | −0.816 | −0.769 | 1.209 | | |
| TopoR | −2.238 | 4.962 | −0.195 | −0.713 | 1.993 | | |

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
