# Peer review of "Testing Different Interpolation Methods Based on Single Beam Echosounder River Surveying. Case Study: Siret River"

_ijgi, doi:10.3390/ijgi8110507_

Round 1
Reviewer 1 Report
The article describes the tests of four interpolation methods for creating bathymetric map. The proposed method seems to be very interesting, but the article does not show its potential. The article contains many errors that should be considered when writing scientific publications at a global level. The title of the article contains the concept of bathymetric map. How does the bathymetric map relate to the content of the article? In the abstract, the authors mention the importance of bathymetric measurements, e.g. in erosion rate. How do measurements with SBES relate to this? In addition, in the abstract, the authors mention the creation of surfaces for hydrologic analysis. What kind of analyses are these? I think that the authors did not properly analyse the literature when writing the article. Why the authors do not mention the use of artificial neural networks, especially deep learning? What is missing here is the correct analysis of the literature which introduces the problem of research work. There is no clear purpose for this research. What are bathymetric maps and what will they be used for? The authors wrote “The best way to collect precise data for a river is to survey it in a specific way, namely transversal cross-sectional paths”. Where do the authors have such a conclusion from? There is no such statement in the literature cited. In addition, the authors wrote that “The spatial resolution of collected data can be improved either by using a very expensive echosounder or by applying different methods of spatial interpolation in a post-processing step.” When creating bathymetric maps, this approach is unacceptable. It is possible that the authors have forgotten to add the appropriate literature or misrepresent their thoughts. There is no clear reason why the authors chose only these four methods of interpolation. The given source is missing for formula (1). Is this the author's formula? One of my main questions that arises is why the authors did not use the test data? Tests should be performed on test data and then on real data. The authors wrote “During the bathymetric measurements, data processing, and production of bathymetric maps 140 we followed International Hydrographic Organization (IHO) S-44 regulation”. Where in the s-44 standard is written about map production? The authors mention that “this particular GIS software has been used to create digital soil mapping, generated by Kriging interpolation, in different urban areas [27]. Also, the GIS software was used by de Amorim Borges [28] for comparing spatial interpolation methods to estimate the precipitation distribution.” How does this relate to the content of the article? Description of the TopoToRaster method is incomprehensible. the authors mention the coefficients spline polynomial functions of degree m and a class of smoothness k. Formulas is missing. The models in Figure 6 should have the same colour scale. This may be misleading, e.g. red on one depicts a depth of 8.36 and on the other already 8.57. Analysing Figure 6 I can only say that the TopoToRaster method smoothes the bottom model. Authors wrote “ TopoR DEM has a much smoother and flatter graphic representation, with lower sinuosity elements.” How does this relate to the precision of the method? I think that the use of 5 profiles is not sufficient. The authors also did not avoid editing errors. For example, on page 10 he wrote “The differences between each model and measurements for each cross section is shown in figure 10.” And then on page 11 they wrote “The differences between SBES measurements and interpolated elevation value, for each of the five section, are shown in Figure 10.” What does the term "hydrologically correct DEM" mean? Maybe it has to do with the lack of a clearly defined research goal? The title is bathymetric map and here "hydrologically DEM". All parameters mentioned by the authors in conclusions should be included in this manuscript. Not in a future article. Figure A1 lacks a legend for the TopoR method. The authors chose 5x5 resolution of DEM. This is too high resolution. It is impossible to make a bathymetric map for such resolution. When using SBES measurements, the smallest resolution should be used. In the case of large data sets (which is very common in practice), it is enough to divide it into smaller subsets. Smaller sets will not cause long computation time. It is proposed to improve it by considering all the suggestions of the reviewer and editing the entire text, including adding tests on test data and reducing resolution.
Reviewer 2 Report
The article is very interesting and needed. Currently, many interpolation methods are used, often without knowing which of them is the best. The article answers the questions which of them give the results with the lowest error.
Regarding the methodology used and its description, the reviewer makes no comments. Noteworthy is the description of the algorithms used in the most popular interpolation methods. The article is also very well written in English. However, I would like to ask you to respond to the following:
1. Models developed using various interpolation methods were obtained from SBES points . The text shows that the resulting models were then referred to the same points. Did the authors consider carrying out the cross-validation process (i.e. generate models from some points, and check those models from another part).
2. It is a great pity that the authors did not check models with other parameters (eg with a resolution of 1 on 1 m or 10 on 10 m).
3. A certain limitation of the research is the fact that the interpolation process was checked only for bathymetric measurements. Whether it is planned to extend the scope of research to other areas measured data - this would increase the universality of the tests carried out to other areas of application of interpolation algorithms.
4. The article is very well written. When it comes to editing errors, it seems to me that on formula 5 is missing one parenthesis.
Reviewer 3 Report
The article corresponds to the profile of the journal and concerns important issues related to geo-information. Detailed solutions to spatial interpolation problems have already been widely published in the literature. It is worth noting, however, that interpolation is applied to river bathymetry. This is a novelty and deserves recognition.
I have only a few minor observations.
Mean SD was used to assess the accoracy of interpolation. In accurate interpolation (e.g. IDW) there are no differences between the measured and unmeasured interpolated values. One can only guess that these differences were determined by cross-validation. It is worth noting.
Among many kriging methods, simple kriging and universal kriging were used. However, ordinary kriging (in my opinion the most commonly used) was not used. I would like to ask for a short justification.
The theoretical model of the semivariogram is very important for interpolation by krigging. Please explain which model of the semivariogram was adopted for interpolation.
Round 2
Reviewer 1 Report
I still maintain the following comments: The title of the article contains the concept of bathymetric map. How does the bathymetric map relate to the content of the article? I think that the use of 5 profiles for local validation of each DEM is not sufficient. The authors chose 5x5 resolution of DEM. This is too high resolution. It is impossible to make a bathymetric map for such resolutionAuthor Response
Please see the attachment

This manuscript is a resubmission of an earlier submission. The following is a list of the peer review reports and author responses from that submission.
Round 1
Reviewer 1 Report
The manuscript describes the performance of interpolation methods in creating accurate bathymetric maps based on SBES surveying. After reading the article, I still think that the title itself is inconsistent with the content of the article. The title is clearly shown “accurate bathymetric map” and there is nothing about it in its content. Only one sentence was added to the abstract which is insufficient.
The authors should take into reconsideration what follows:
1. in the abstract, GIS was mentioned; nothing is about it in the article; a few lines have been added about ArcGIS software; ArcGIS software is not GIS itself; ArcGIS is only a tool used to create GIS
2. in the abstract, precise maps were mentioned; the use of SBES precludes the preparation of a precision map; it still has not been improved; you can not make a precise map from the data received from SBES; unless clearly representative what this precise map is; especially that the authors later say that “he sound velocity in water ranges between 1400 to 98 1600m/sec, but, when a sound velocity profiler (SVP) is not available, the echosounder uses a 1500 99 m/sec average value”
3. what regulations were taken into account during hydrographic measurements (international or national)? – standard S-44 should me added to references
4. What is the resolution of the created DEM? Is it the same at every stage of development and for each method? It was mentioned that at one of the stages it is 5x5 m. It is too big for the river area (especially at such a small width) - I still maintain my opinion
a. In case the authors write about accurate maps, it is not possible to apply such a high resolution. For research purposes, the tested area can be divided into smaller areas. The grid itself is not a very heavy product. Only its creation can take a while. In this case, the resolution can be 1m.
5. the description of attached figures and tables is missing - the reviewer was concerned that every figure and table should be described in the content of the article. It is not enough to mention them. They should be clearly described.
6. it is a big mistake to write that “Our study points out that 281 the measurements combination in a more uniform grid (combination between transversal and 282 longitudinal bathymetric surveys) can provide a higher precision to the data processing”. This statement does not have to be confirmed by tests - the added phrase (line 387-390) is an obvious thing and it does not seem that you need to research it.
Additionally, authors have not given a detailed enough overview of hydrographic data processing field. The latest research focuses on the application of artificial neural network, especially deep learning in this area. Some key papers were not mentioned, which are dealing with the using of ANN, such as:
• Stateczny, A.: The neural method of sea bottom shape modelling for the spatial maritime information system.
• S. Liu, L. Wang, H. Liu, H. Su, X. Li and W. Zheng, Deriving Bathymetry From Optical Images With a Localized Neural Network Algorithm
• Wlodarczyk-Sielicka, M.; Lubczonek, J. The Use of an Artificial Neural Network to Process Hydrographic Big Data during Surface Modeling
• Ghorbanidehno, H.; Lee, J. H.; Farthing, M.; Hesser, T.; Kitanidis, P. K.; Darve, E. F. Bathymetry estimation using deep learning techniques
• Lubczonek J., Wlodarczyk-Sielicka M. The Use of an Artificial Neural Network for a Sea Bottom Modelling
You should consider to resign from “bed elevation” to “depth”.
The table 2 was separated in Table 1 and Table 2. Where is table 1 now (the old one)? besides what does T-stat, P (T <= t) two tail and P-value mean?
The authors wrote that “We investigate here the 59 performance of four different interpolation techniques by applying different statistical methods of 60 performance assessment, such as minim value (MinV), maxim value (MaxV), mean error (ME), mean 61 absolute error (MAE), mean square error (MSE), root mean square error (RMSE), root mean square 62 standardized error (RMSSE), standard deviation (SD), coefficient Kurtosis (Kurt), Skewness (Skew)”. Where in the article will we find all these values?
On line 291 it was written that “models and measurements are presented in table A1
292 (Appendix B)”. It should be table B.1. In addition, the results contained therein are not discussed in the text.
Reviewer 2 Report
In this manuscript, the authors use a section of the Siret River in Romania to examine and compare 4 methods of interpolation from single-beam data to create a comprehensive map of river bathy. The study is neither particularly novel nor extraordinarily well constrained. The interpolation methods are some standard ones, available in the ArcGIS package and this study is somewhat dependent on that software. The choice of which 4 interpolation methods to be used seems based on the default choices in the ArcGIS software rather than any overall scientific justification.
My main problem with this article is that the authors use the SBES data for the interpolation and for testing the interpolation. It would be far more compelling and worthy of publication if the river had a multibeam or LiDAR survey as independent baseline data to compare against their data and interpolation. A major secondary problem is that only 5 profiles are used to compare the interpolation to "ground truth" and it's unclear if the 5 profiles contain data used in the interpolation or not. What are these testing data? They are not adequately described, only on lines 280-283. More than 5 profiles should be used however (Figure 9) and none on a really straight reach (Figure 11). Third, it's hardly surprising from a geostatistics point-of-view that this study (and others cited by in this work itself) that ANUDEM, or the TopoR version used herein, produces the most accurate interpolation because it has the most degrees of freedom. TopoR is essentially just the spline interpolation with a smoothing function added. It's also not ideally implemented here because the ancillary data that it's really designed to incorporate (lines 219-220) are not available in this study.
Throughout section 2, several functions and equations are presented. Each interpolation method as specific parameters used for tuning. None of these are given where I could find them. It's important to know, for example, not just what kriging functions were used but also what the semivariograms are that were used to construct the model. As it is written, I have no confidence that the kriging was done optimally. If the point is that default ArcGIS kriging may not be the best, then I agree, but that's pretty well known. There's no discussion of how ANUDEM via TopoR was specifically implemented. This is necessary to include in the Ancillary Material at least. The most important, and really the main cause of the differences in interpolation, based on the evidence presented is the channel edge effects. Interpolators that perform poorly in high-gradient environments are probably just using default values from some software. Is that the case here? I've had luck in tuning the interpolation parameters to get better results if that's the case.
Nevertheless, this was an interesting study to read. The statistical analysis is good, and while this is not ground-breaking science, it confirms some previous work on generating digital terrain models from sparse-clustered point cloud data. With more rigorous data collection and more detailed explanation of specific application of the methods, it might be more interesting.